# Industry involvement in evidence production for genomic medicine: A bibliometric and funding analysis of decision impact studies

Gillian Parker[1], Sarah Hunter[1], Stuart Hogarth[2], Fiona A. Miller[1] *

1 Institute of Health Policy, Management and Evaluation, University of Toronto, Toronto, Ontario, Canada,
2 Department of Sociology, University of Cambridge, Cambridge, United Kingdom

* fiona.miller@utoronto.ca

**Data Availability Statement:** All relevant data are within the paper and its Supporting Information files.

## Abstract

### Background

Decision impact studies have become increasingly prevalent in genomic medicine, particularly in cancer research. Such studies are designed to provide evidence of clinical utility for genomic tests by evaluating their impact on clinical decision-making. This paper offers insights into understanding of the origins and intentions of these studies through an analysis of the actors and institutions responsible for the production of this new type of evidence.

### Methods

We conducted bibliometric and funding analyses of decision impact studies in genomic medicine research. We searched databases from inception to June 2022. The datasets used were primarily from Web of Science. Biblioshiny, additional R-based applications, and Microsoft Excel were used for publication, co-authorship and co-word analyses.

### Results

163 publications were included for the bibliometric analysis; a subset of 125 studies were included for the funding analysis. Included publications started in 2010 and increased steadily over time. Decision impact studies were primarily produced for proprietary genomic assays for use in cancer care. The author and affiliate analyses reveal that these studies were produced by 'invisible colleges' of researchers and industry actors with collaborations focused on producing evidence for proprietary assays. Most authors had an industry affiliation, and the majority of studies were funded by industry. While studies were conducted in 22 countries, the majority had at least one author from the USA.

### Discussion

This study is a critical step in understanding the role of industry in the production of new types of research. Based on the data collected, we conclude that decision impact studies are industry-conceived and -produced evidence. The findings of this study demonstrate the

**Funding:** Funded by an investigator-initiated grant to FAM from the Canadian Institutes of Health Research (PJT 148805) https://cihr-irsc.gc.ca/e/193.html. The funders had no role in study design, data collection and analysis, decision to publish, or preparation of the manuscript.

**Competing interests:** The authors have declared that no competing interests exist.

**Abbreviations:** WoS, Web of Science; CRO, contract research organization.

depth of industry involvement and highlight a need for further research into the use of these studies in decision-making for coverage and reimbursement.

## Background

The use of genomic tests to guide treatment decision-making has grown significantly in recent years [1, 2]. These 'personalized medicine' tools are most prominent in cancer care, but their use to support treatment decision-making for other disease-types is growing [3]. Most of the prognostic multi-marker genomic tests commonly used today are proprietary products, meaning these tests have been conceived of and developed by private sector companies. Indeed, the top five genomic breast cancer prognostics are all proprietary products: Oncotype Dx (Genomic Health/ Exact Sciences), followed by Mammaprint (Agendia), Prosigna (Nanostring), Breast Cancer Index (Biothernostics) and Endopredict (Myriad) [4]. The top prognostic products in other cancer care are, as well, propriety products: Envisia (Veracyte) and Percepta (Veracyte) for lung cancer, and Decipher (Veracyte) for prostate cancer [4]. As these new health technologies enter the market, production of evidence demonstrating their utility in practice is key to supporting their integration in routine clinical practice and to obtaining coverage and reimbursement from private and public payors.

In recent years, a new type of study has been published in the field of genomic medicine research. Decision impact studies report on the impact of a genomic test on decision-making and other aspects of the test's clinical utility [4]. Though new, decision impact studies have already been included as evidence in support of various international health technology assessments [5–8], and have been used in funding decisions for collectively financed health services. While these studies have proliferated in genomic medicine, no synthesis or analysis of this new form of evidence had been generated. This gap in knowledge prompted us to conduct a scoping review of decision impact studies for genomic assays in cancer care to begin to understand and interrogate this new form of evidence [4]. The scoping review found that decision impact studies in genomic medicine report on a range of potential indicators of clinical utility, primarily for proprietary genomic tests. Thus, decision impact studies appear to be positioned to provide evidence in support of securing coverage and reimbursement. These results provided compelling insights, but also revealed a gap in understanding industry's involvement in the production of decision impact studies and how these studies were funded. Therefore, we decided to conduct bibliometric and funding analyses to further explicate the characteristics and motivations of these studies and to expand the knowledge base on this emerging type of evidence in genomic medicine.

Understanding industry involvement in scientific evidence production is an evolving and increasingly complex endeavour. In recent years a growing body of scholarship has explicated industry's role in evidence production, drawing attention to the opacity and self-interested nature of this involvement [9–14]. The majority of focused research on this topic has analyzed the influence of industry on evidence production for tobacco [15], alcohol [11, 12, 16] and pharmaceuticals [9], with some research on medical devices [10], but little focus on health technologies used in genomic medicine.

A key aspect of industry involvement in scientific evidence production relates to industry's role in funding academic research and financially supporting academic researchers. Scholars have raised concerns regarding the impact and influence of industry funders on the rigour and objectivity of scientific outputs [9, 11]. Álvarez-Bornstein and colleagues argue that all funding bodies influence research communications, decision-making and the type of knowledge

produced, but raise particular concerns about the impact of private sector funding on bias to positive outcomes and lack of prioritization of public good in health research [17]. Authors have also noted that when a particular research study is guided by commercial imperatives, the research agenda is likely to be altered [16]. Despite the importance of transparency on these factors, scholars have reported that funding information and 'conflict of interest' disclosures, which are intended to increase transparency of industry involvement and funding, are often missing, incomplete or inaccurate in biomedical and other scientific publications [9, 11, 17–19]. Researchers have begun to explore the 'reward triangle' of authorship, citations and acknowledgments as a way to explicate the key factors influencing research impact [20, 21]. While this approach has been criticized [17], it draws attention to the third and often over-looked aspect of research impact—funding and disclosures—that, with the increase of private involvement in health research, is critical to understanding and evaluating to motivations and intended purpose of new evidence production.

Another key aspect of this new knowledge production concerns the prolific, but often 'invisible' relationships between industry and researchers in the production of evidence, which support the commercial success of health technologies [22–25]. There have long been 'invisible communities' [26, 27] or 'invisible colleges' [28–30] of influential researchers, who foster knowledge creation through collaboration networks, but whose activities are not made explicit. More recently Demortain has shown that such 'invisible colleges' play a critical role in generating knowledge for regulatory purposes, becoming authorized as 'regulatory science' for the purposes of regulation of risk and standardization of control [31]. Holloway, Miller and Simms extended this conceptualization to explicate the influence of the invisible college in the context of the regulation of genomic medicine, influencing knowledge standards and norms related to clinical guidelines and coverage and reimbursement decision-making [25].

Through these processes, a new type of corporate science is becoming influential in health policy. Though made to, "look like traditional academic work, . . . [it is] performed largely to market products" [9(p.171)]. In addition to author collaboration networks, scholars have raised concerns about how industry is increasingly being embedded throughout the research process [14, 32]. Sismondo highlighted the significance of 'publication plans', through which industry endeavours to, "extract[s] the maximum amount of scientific and commercial value out of data" [9(p.171)]. These plans often are developed by the industry members and written by independent ghost writers, and involve research conducted by contract research organizations (CROs) [9, 14, 33]. Researchers have raised concerns about the increase in studies conducted through CROs and implications for objectivity and rigour [14, 34]. Guo and colleagues report that industry involvement and funding of clinical trials has led to a diversion of funding from regulated, academic arrangements to for-profit CROs [14].

While important, the research to-date on industry involvement in scientific evidence production has primarily focused on the impact of industry funding on bias, reporting of favourable results [11, 13, 14] or research-agenda-setting [35]. Industry's role in generating evidence to expedite health technology coverage and reimbursement has just begun to be explored in the literature [25, 36]. This paper aims to extend this literature by exploring industry's role in evidence production for decision impact studies in genomic medicine through both a bibliometric analysis and a funding analysis. The research questions that motivate this study are: *1. What are the research patterns, trends and key topic areas of decision impact studies? 2. Who/ what are the prominent authors, collaborations institutions, funders and countries in this field? 3. What is the depth and range of industry involvement in decision impact studies?* The objectives of this study were to: (a) map the evolution of decision impact studies in genomic medicine, and (b) identify and analyze the scope of industry involvement in these studies including, authors, institutions, affiliates, networks, and funding.

## Methods

A bibliometric analysis is a distinct type of literature synthesis that analyses the aggregated 'bibliometric metadata' associated with a particular set of publications [30, 37, 38]. Bibliometric metadata comprises the data associated with a publication, generated at the time of publishing (e.g., publication authors, author affiliations, keywords, cited references, or total citation count) [30, 37, 38]. A funding analysis is a type of bibliometric analysis, as it also employs descriptive statistical techniques to analyze bibliometric metadata associated with a particular set of publications. By employing descriptive statistical techniques, key topic areas within a field of study can be identified, and the influence, spread and emerging areas of a body of research can be examined, often through software-generated data visualizations [37]. An earlier version of the protocol was registered with Open Science Framework: osf.io/hm3jr.

### Search strategy

Publications for inclusion in our bibliometric and funding analyses were systematically identified using a search strategy developed for our previously conducted scoping review focused on decision impact studies in genomic medicine research [4] (see also S1 Appendix). Four databases that were expected to return publications most relevant to decision impact studies in genomic medicine were searched: Medline, Embase, Scopus, and Web of Science (WoS). The search string used included: 'decision impact' or 'decision-impact' or 'decision-making impact' or 'decision making impact'. For each database, the search was limited to peer-reviewed publications, including articles and conference abstracts. As the focus of the study was the production of 'scientific' evidence, gray literature was excluded. Due to resource constraints, only English-language publications were included.

### Selection of publications for datasets

The results from our database searches were imported into Covidence, a Cochrane technology platform (www.covidence.org), in order to facilitate the selection of publications for the bibliometric and funding datasets. The same procedures were used to select relevant publications for both the bibliometric and funding components of the study during the title and abstract screening-stage. Procedures began to diverge at the full-text review-stage. Inclusion and exclusion criteria were established at the outset in order to facilitate these step-wise publication selection processes.

**Title and abstract screening.** Two researchers (Ghazi and Hunter) independently reviewed the titles and abstracts of all publications. Publications were excluded if they did not focus on the (i) impact of a genomic-based health technology (e.g., a genomic test) on (ii) clinical decision-making, and in (iii) a health care setting. Scientific journal publications of all types (empirical studies, reviews, commentaries, protocols and conference abstracts) were included. A third reviewer (Parker) checked a random 10% sample of articles to ensure reliability. Discrepancies were discussed and resolved collaboratively.

**Full-text review.** Full-text review is not typical and often not needed for a bibliometric analysis. As the included publications for this study were initially collected for our scoping review, we also report this process in this study. Two researchers (Ghazi and Hunter) independently reviewed the full-text version of each publication that remained following title and abstract screening. Publications were excluded if they were not about a genomic test or tool or not about 'decision impact studies'. The third researcher (Parker) reviewed a random 10% sample of the publications deemed eligible. These publications became the set of publications used to conduct the bibliometric analysis. For the funding analysis dataset, the unit of

observation was a study, so multiple publications of a single study were excluded and only the most recent publication was retained for this sub-analysis.

## Data collection

Different methods were used to collect the bibliometric metadata ('data') for the bibliometric versus funding components of the study.

For the bibliometric analysis: Data for most publications were retrieved from WoS, as recommended by Aria and Cuccurullo [39]. Where data were not listed in WoS, they were located in and manually extracted from other databases (primarily Embase and Scopus).

**For the funding analysis.** Author affiliation, study funding, and disclosure information was manually extracted from each publication. Manual extraction of the data was required because funding metadata is often inconsistently reported and/or categorized in electronic databases [17]. Prior to data collection, a data collection worksheet was developed iteratively, created in Microsoft Excel and piloted with a sample of five publications.

**Data pre-processing.** Data pre-processing included 'data loading and converting' and 'data cleaning' [30, 37]. For the bibliometric analysis, analyses were either automated (using software applications specifically developed to conduct bibliometric analyses) or conducted manually (using Microsoft Excel). Data pre-processing was only relevant to the bibliometric component of the study as the data in electronic databases are often recorded inconsistently or missing [17]. Data cleaning is a critical, yet often overlooked, step in bibliometric analysis. The specific steps taken to first load and convert, then to clean the data were influenced by the particular software applications used. There are a number of software applications available to conduct bibliometric analyses [37]. We chose to primarily use an R-based application, Bibliometrix, because it can be used throughout the entire bibliometric analysis workflow, has an easy-to-use interface (Biblioshiny), has broad capacity for analysis, and has adequate software support available.

**Data loading and converting.** After the available bibliometric metadata were retrieved from WoS, they were loaded into Biblioshiny, which then converted and formatted the data per Bibliometrix requirements. A Bibliometrix-formatted dataset was subsequently downloaded (in a format readable by Microsoft Excel). The data were reviewed, and missing data was manually collected for those publications not found on WoS and manually entered into the Bibliometrix-formatted dataset, according to the existing format of the dataset.

**Data cleaning.** Once the relevant data for all included publications were entered into the dataset, an open-source software, OpenRefine, was used to facilitate data 'cleaning', including the following key items:

- ensuring the consistent spelling and formatting of author names; and,

- ensuring the consistent entry and spelling of author affiliations, and formatting of affiliate addresses.

Such steps are necessary to ensure that the data analysis and/or visualization software used can correctly recognize multiple instances of one author's name, for example, as the same and in turn, correctly tabulate, for example, the number of publications that each author has contributed to. Any changes made were recorded in a data cleaning log.

## Data analysis

Once the data were cleaned and the compatibility of the dataset with Biblioshiny was confirmed, the finalized dataset was loaded into Biblioshiny for analysis and visualization. We used three bibliometric techniques to analyze the data: 'publication analysis' (for the

bibliometric and funding components), 'co-authorship analysis' (for the bibliometric component), and 'co-word analysis' (for the bibliometric component). All metadata used to conduct the analyses are included in the appendices (see S2 and S3 Appendices).

*Publication analysis* examines the "frequencies (numbers) and patterns related to the act of publication" [40 (p.99)]. For the bibliometric component, publication analyses relied on descriptive statistics (counts) and were carried out automatically using the R-based application, Biblioshiny, and manually using Microsoft Excel. For the funding component, publication analyses also relied on descriptive statistics (counts) and were carried out manually using Microsoft Excel.

*Co-authorship analysis* enables a researcher to examine the social networks of a particular field of research [30]. When two or more authors publish together, a relationship is established; co-authorship is considered to be a measure of collaboration demonstrated at either institution- or country-level [30]. For the bibliometric component, co-authorship analyses relied on network analysis [30] and were carried out automatically using the R-based application, Biblioshiny.

*Co-word analysis*, unlike other bibliometric analytic techniques, examines the content of publications (primarily: publication titles, abstracts, or keywords) in order to identify concepts or topics relevant to a particular field of research [30]. Phrases that frequently appear in a particular set of publications are considered to reflect the prominent concepts or topics within that field of research [30]. For the bibliometric component, the co-word analysis was carried out automatically using the R-based application, 'wordcloud' [41]. The code used to conduct this analysis has been included in the appendices (see S4 Appendix).

## Data visualizations

Data visualizations were generated automatically (through the R-based applications Biblioshiny or wordcloud) and then reproduced (using Adobe Illustrator) to provide clarification or additional analysis. Upon review of software-generated visualization, the research team determined that the 'raw' visualizations required additional analysis to comprehensively interpret and display the results.

## Results

### Literature search

In total, 2,110 publications were returned from the four databases searched; 1,803 publications remained after duplicates were removed; 269 publications remained following the title and abstract screening stage. Following the full-text review stage, 163 publications (65 articles, 82 conference abstracts, 16 commentaries or opinions and reviews) were included in our set of publications for the bibliometric analysis, and a subset of 125 publications (62 articles, 50 conference abstracts, 13 reviews) reporting on unique studies were included in our set of publications for the funding analysis. See Fig 1 for the flow diagram outlining the study design. See S1 Table for a table of included publications.

### Bibliometric analysis

**Publications over time.**   All included publications were published between 2010 and June 2022. Fig 2 compares production over time of: (a) publication-types—articles versus conference abstracts versus commentaries or opinions and reviews; and (b) publication-focus—breast cancer versus other cancers versus other diseases. In terms of publication-type: conference abstracts (n = 82) represented 50% of the sample; articles (n = 65) represented 40%; and

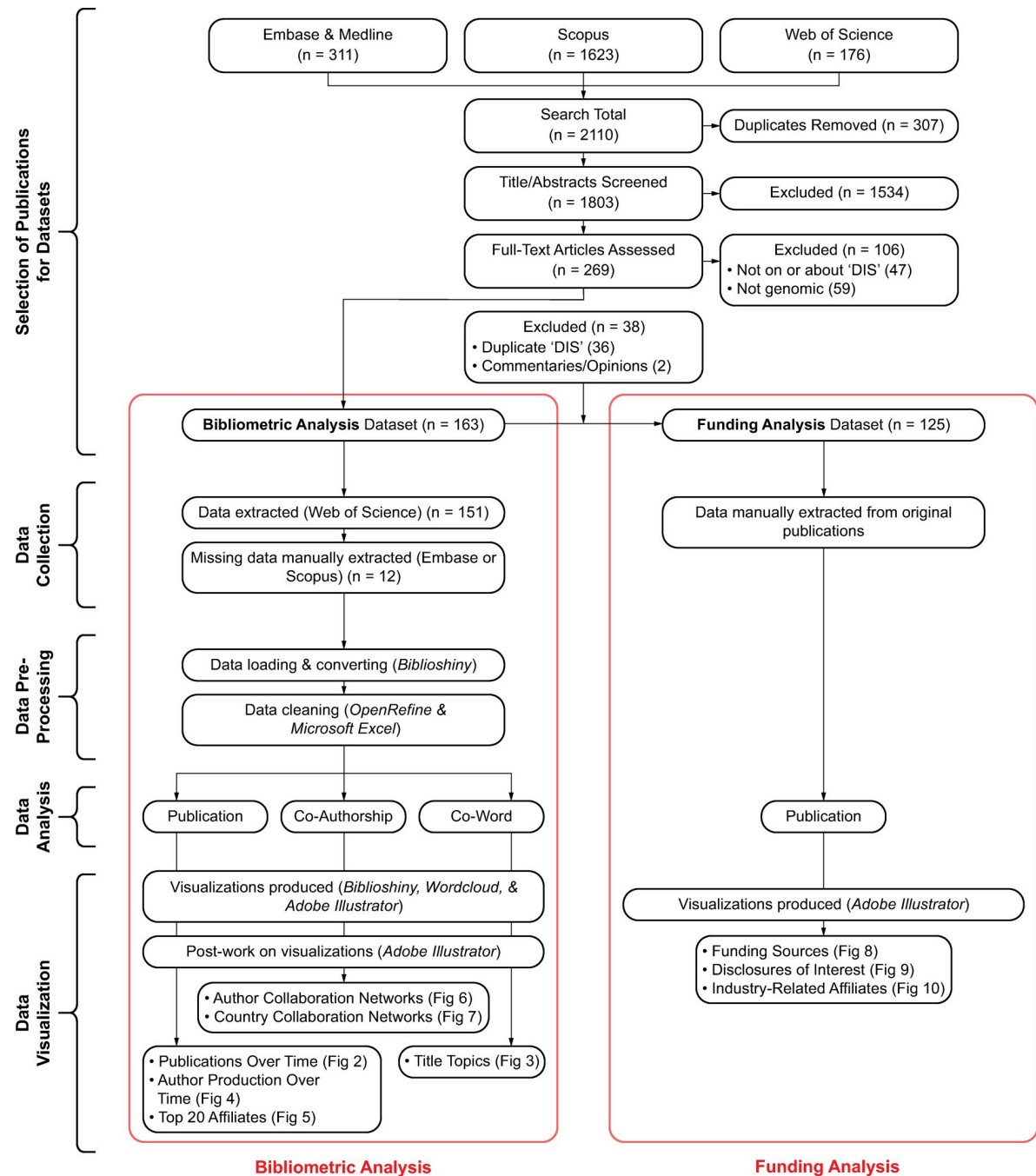

**Fig 1. Flow diagram outlining the study design.**

commentaries or opinions and reviews (n = 16) represented 10%. Publications primarily focused on breast cancer (n = 122), other cancers (n = 35), and then other diseases, such as lung disease and rheumatoid arthritis (n = 6). Publications focused on other cancers first appeared in 2011 and other diseases, in 2021. Publications reporting on or about 'decision impact studies' peaked in 2016 (n = 21). Sixteen publications were published during the first six months of 2022. Twelve out of the 16 publications published in 2022 were focused on breast cancers; two were focused on other cancers; and the remaining two, on other diseases.

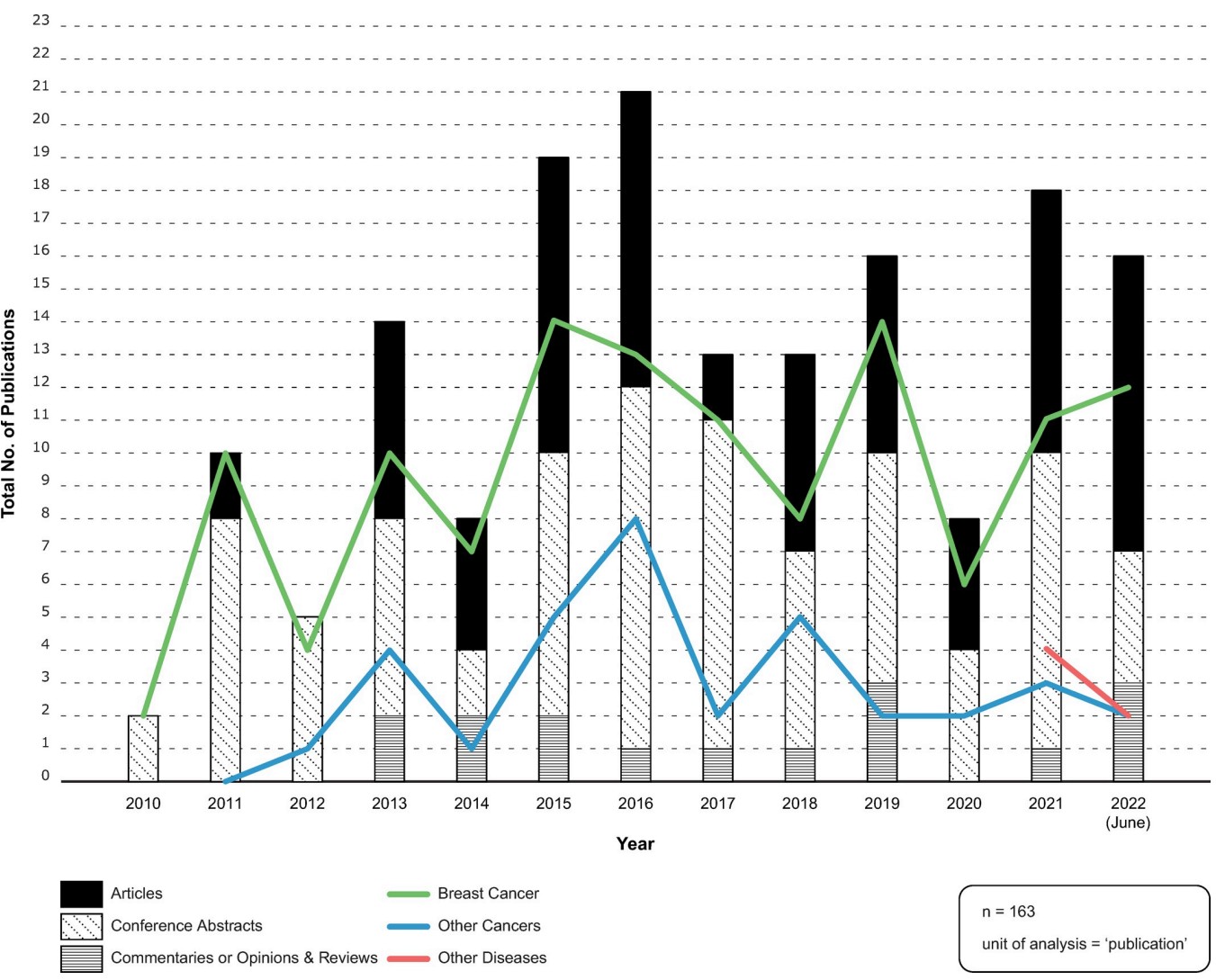

**Fig 2. Publications over time.** Source: Microsoft Excel: analysis; Adobe Illustrator: visualization.

### Title topics

Fig 3 reflects the 'bi-grams' (i.e., two-term phrases) that appeared most frequently (five or greater times) within publication titles. These bi-grams can be categorized into five distinct themes: (1) cancer-related; (2) impact- or outcomes-related; (3) proprietary- or industry-related; (4) genomic-related; and (5) methodology-related. The bi-grams relevant to each theme are indicated by colour. Of particular note, the cancer-related bi-gram 'breast cancer' appears most frequently within publication titles (n = 104). Further, the proprietary- or industry-related bi-grams that appeared five or more times in publication titles included 'Recurrence Score' (n = 33) and 'Oncotype Dx' (n = 31); the latter is the name of the breast cancer prognostic genomic test developed by Genomic Health [42]. 'Recurrence Score' is the score provided by Oncotype Dx, based on the assessment of the expression of 21 genes [43].

### Top 20 author production over time

Fig 4 depicts the top 20 producing authors (i.e., authors that had published the most during the publication timeframe, 2010–2022). The size of the circle indicates the number of

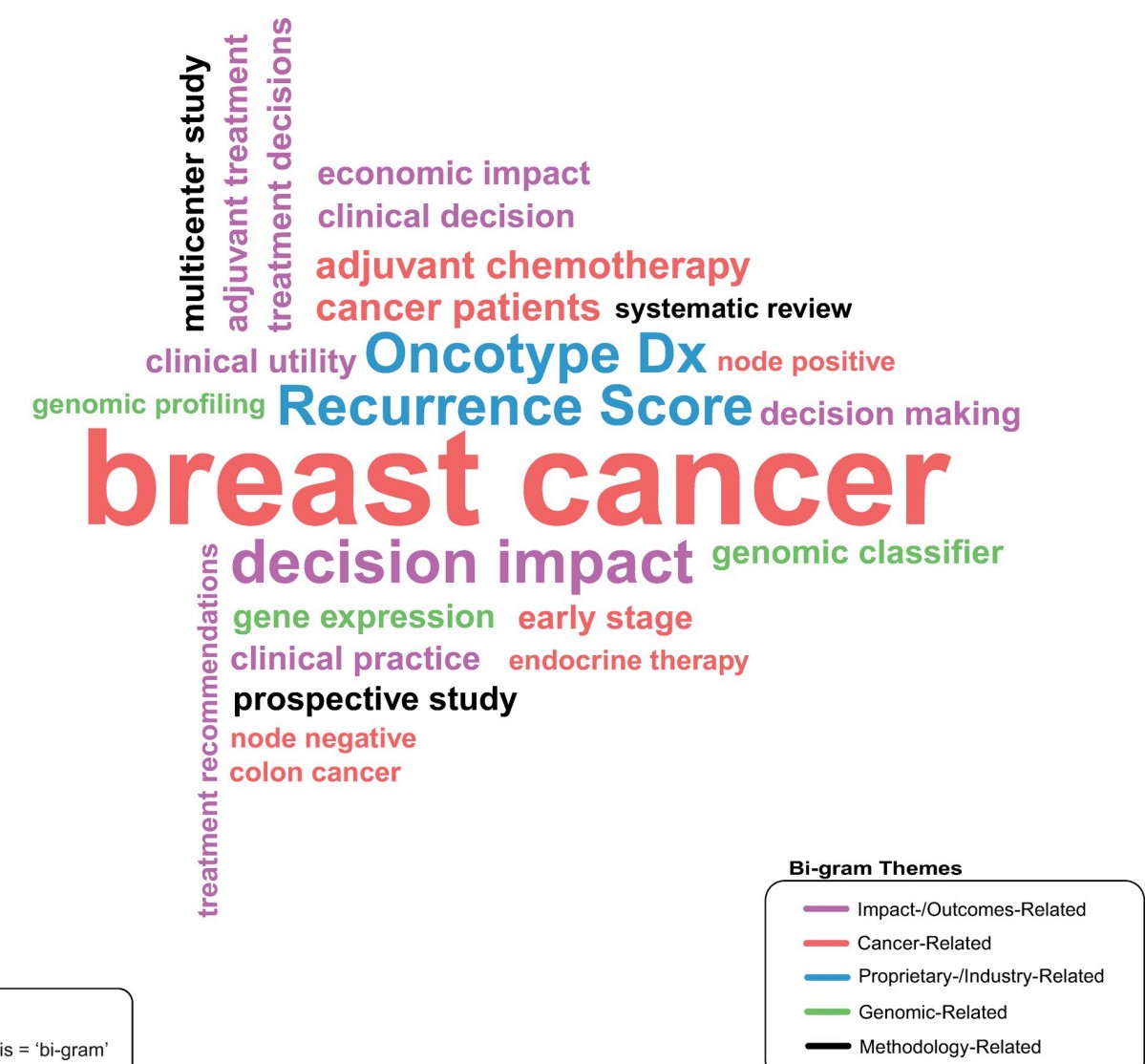

**Fig 3. Title topics.** Source: R-based application 'wordcloud' was used to generate the word cloud; additional analysis (i.e., categorization of the bi-grams) was completed by the study authors and applied to the visualization using Adobe Illustrator. Parameters specified: n-gram-type (phrase-length) = 'bi-grams'; minimum n-gram frequency = 5.

publications an author had published in a particular year; a blue ring around an author's name indicates that the author was USA-based (four of top 20 authors), while a gray ring indicates that the author was Europe-based (16 of top 20 authors). Authors appeared to begin publishing during two prominent timeframes: nine authors began publishing in either 2011 or 2012; and another nine authors began publishing in 2015 or 2016. Three of the top 20 producing authors have published during the first six months of 2022.

## Top 20 affiliates

In total, 924 authors were responsible for producing the 163 publications included in the dataset used for the bibliometric analysis. Fig 5 depicts the top 20 author affiliations. Six of the top

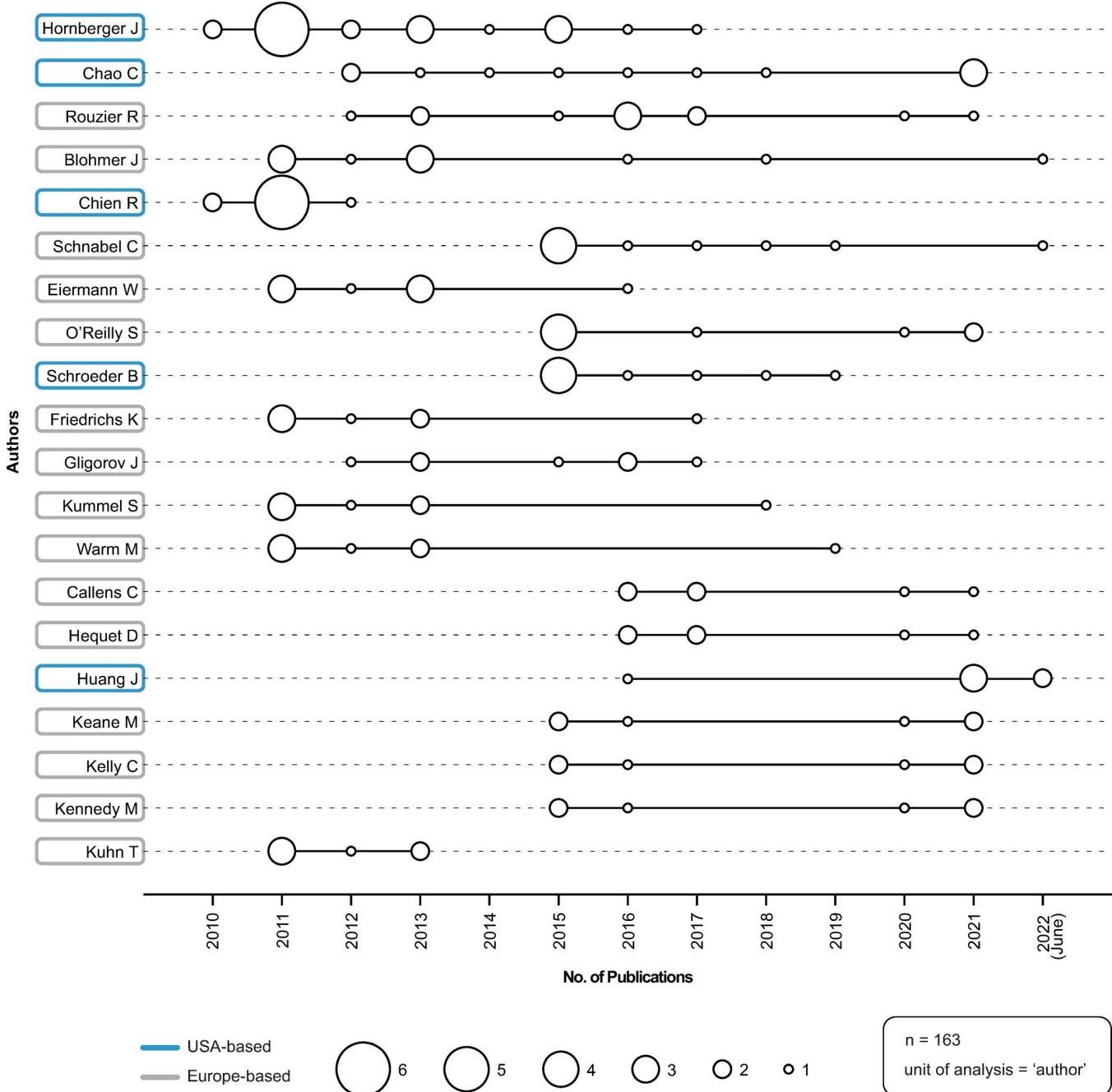

**Fig 4. Top 20 author production over time.** Source: R-based application Biblioshiny was used; additional analysis (i.e., affiliate region of origin) was completed by the study authors and applied to the visualization using Adobe Illustrator. Parameters specified: number of authors = 20.

20 affiliates were industry-related (red ring)—and these were all based in the USA. Of note, the combination of Genomic Health and Exact Sciences (Genomic Health was purchased by Exact Sciences in 2019 [42]), would constitute the top industry affiliate (n = 20). The remaining 14 affiliates were medical- or academic-related (gray ring). Of these, six were based in Germany, three in France, two in USA, two in Ireland, and one in Israel.

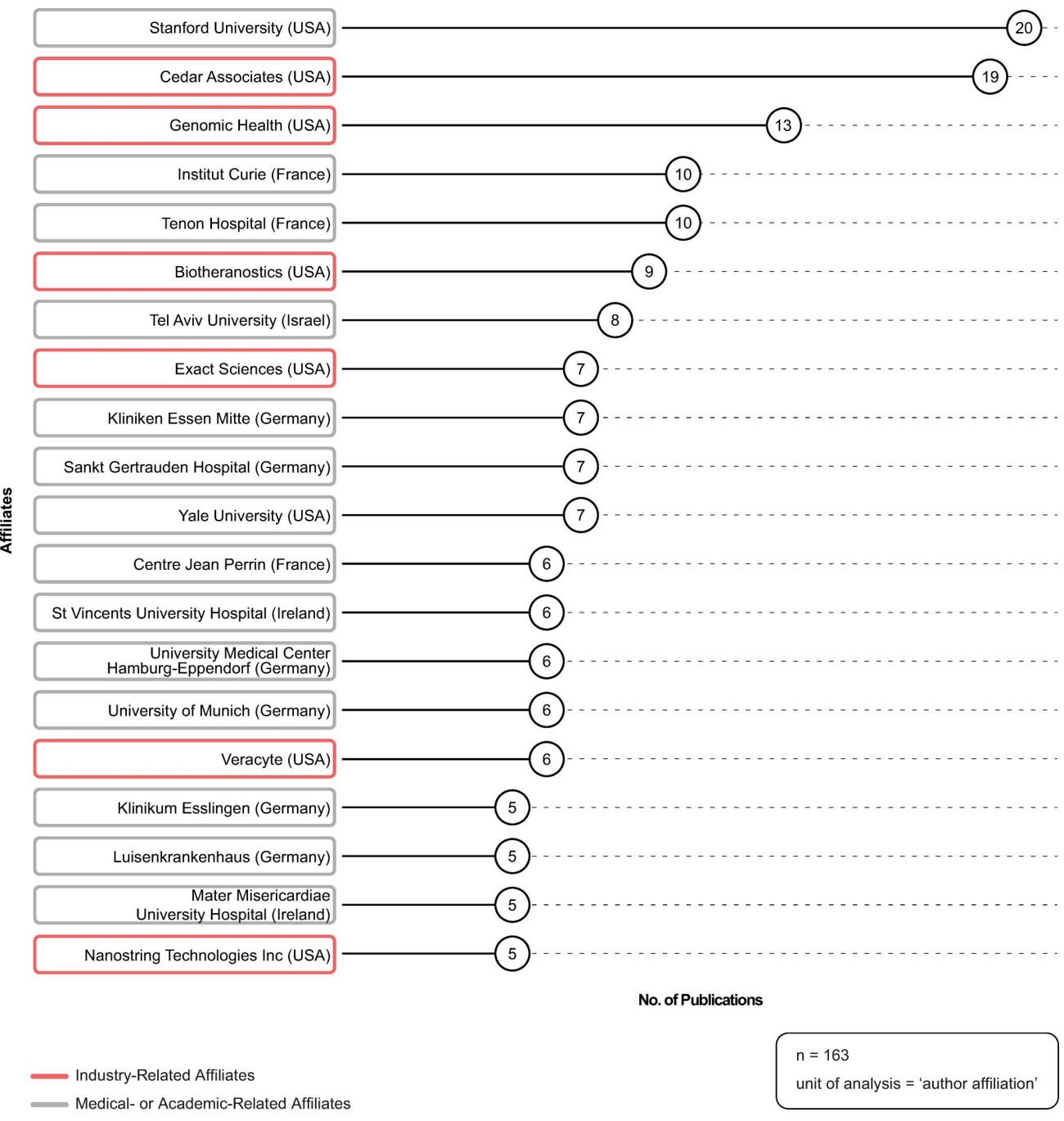

**Fig 5. Top 20 affiliates.** Source: R-based application Biblioshiny was used; additional analysis (i.e., affiliate-type, industry- versus medical- or academic-related) was completed by the study authors and applied to the visualization using Adobe Illustrator. Parameters specified: affiliation name disambiguation = no; number of affiliations = 20.

## Author collaboration networks

Fig 6 depicts the publication collaboration networks between authors. Each circle, or 'node', represents one author. Each line connecting two nodes represents at least one publication collaboration between authors; the shorter the line, the stronger the publication collaboration between authors (i.e., the more shared publications). Publication collaboration networks were

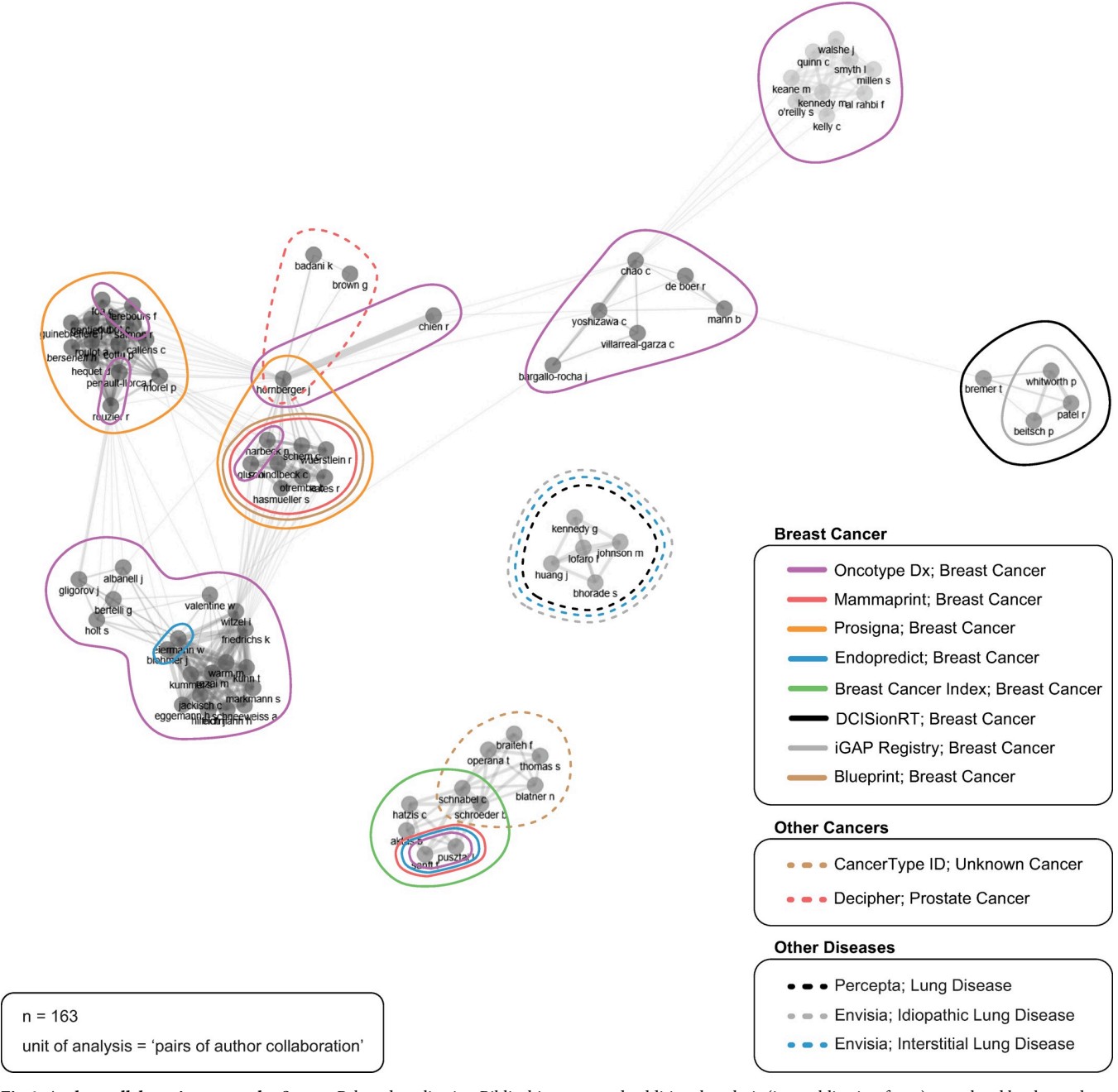

**Fig 6. Author collaboration networks.** Source: R-based application Biblioshiny was used; additional analysis (i.e., publication focus) completed by the study authors and applied to the visualization using Adobe Illustrator. Parameters specified: network layout = automatic layout (default); clustering algorithm = walktrap (default); normalization = association (default); number of nodes = 92 (top 10% of total authors); repulsion force = 0.1 (default); remove isolated nodes = yes (default); minimum number of edges (i.e., connections) = 1 (default).

focused on research surrounding particular genomic tests and particular types of cancers or other diseases. The majority of publication collaboration networks (n = 18) were focused on genomic tests used in breast cancer care, followed by two focused on genomic tests used in other types of cancer care and three focused on genomic tests used for other diseases (lung diseases: general, idiopathic, and/or interstitial). Together, 13 different proprietary tests were represented by these networks.

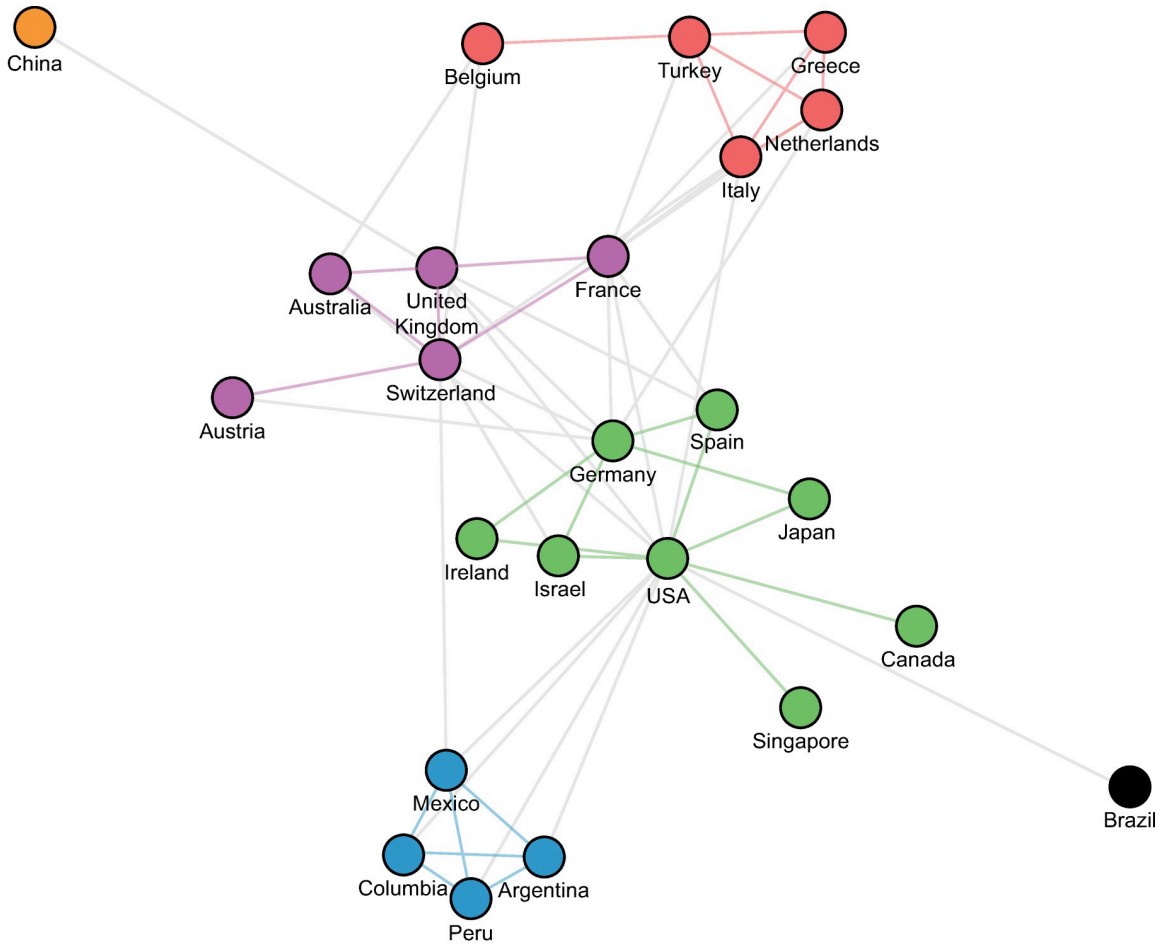

Note: Distinct collaboration networks identified
       by node colour.

n = 163

unit of analysis = 'pairs of country collaboration'

**Fig 7. Country collaboration networks.** Source: R-based application Biblioshiny was used; visualization was recreated using Adobe Illustrator. Parameters specified: network layout = automatic layout (default); clustering algorithm = walktrap (default); normalization = association (default); number of nodes = 2000; repulsion force = 0.1 (default); remove isolated nodes = yes (default); minimum number of edges (i.e., connections) = 1 (default).

### Country collaboration networks

Fig 7 depicts the publication collaboration networks between countries (i.e., the countries of origin of authors' affiliations). Each circle, or 'node', represents one country. Each line connecting two nodes represents at least one publication collaboration between countries. Like the publication collaboration networks between authors, the shorter the line, the stronger the publication collaboration between countries. The coloured nodes and lines represent authors from countries that publish together more frequently, and the gray lines represent weak connections between networks. Four primary publication collaboration networks between countries were identified (blue, green, purple, and red). Of note is the centrality of the USA within the green network and between all networks. Table 1 Identifies the country of origin of the top 20 affiliates.

**Table 1. Top 20 affiliates, country of origin.**

| Percent (%) | Top 20 Affiliates by Country |
|---|---|
| 40 | USA |
| 30 | Germany |
| 15 | France |
| 10 | Ireland |
| 5 | Israel |

## Funding analysis

**Funding sources.**   Fig 8 depicts the funding sources associated with the publications reporting or reviewing empirical results of decision impact studies undertaken in genomic medicine research. Sixty-one publications explicitly reported that they received funding from one or more funding sources. Fifty-six publications did not report any information about funding. Eight publications reported that they did not receive any funding. Of those 61 publications that explicitly reported that they received funding, the majority of the funding sources were industry-related (n = 56).

## Disclosures of interest

Fig 9 depicts the industry-related disclosures of interest reported in the publications. Over half of the publications (n = 71) explicitly reported one or more industry-related disclosures, including: employment with, stock ownership in or advisor to a particular industry-player; financial, statistical or editorial support received from a particular industry-player; and in-kind contributions (e.g., donation of genomic test technology) by a particular industry-player. Thirty-five publications did not report any information about disclosures of interest.

## Industry-related affiliations

Fig 10 depicts ratios of industry-related and non-industry-related author affiliations associated with the publications. Seventy publications explicitly reported one or more industry-related author affiliations. Fifty-four studies did not report any industry-related author affiliations.

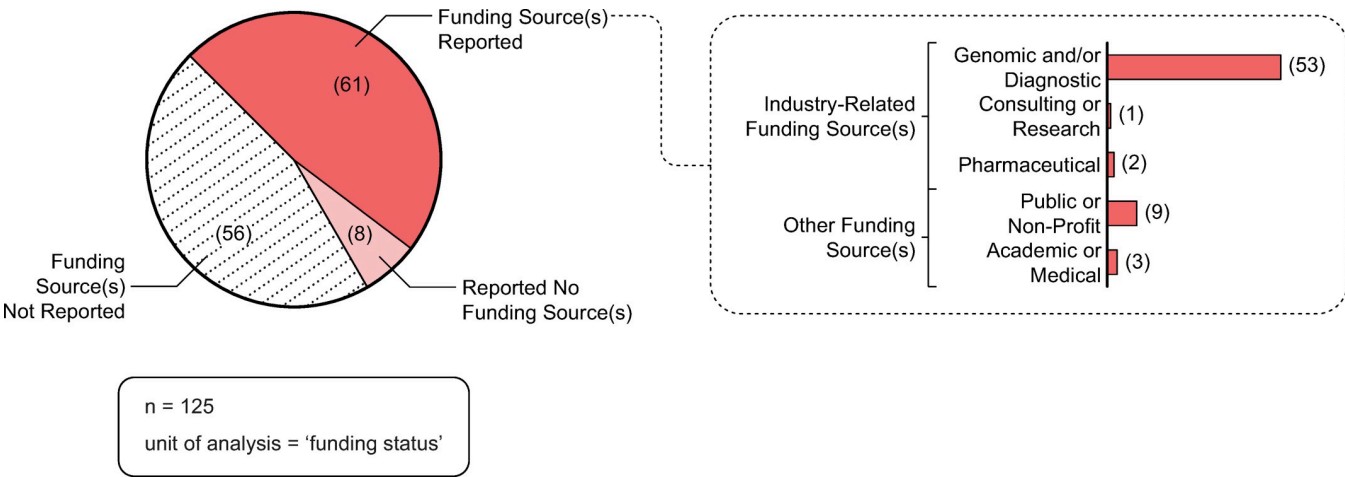

**Fig 8. Funding sources.** Source: Microsoft Excel was used to facilitate data analysis; Adobe Illustrator was used to facilitate data visualization.

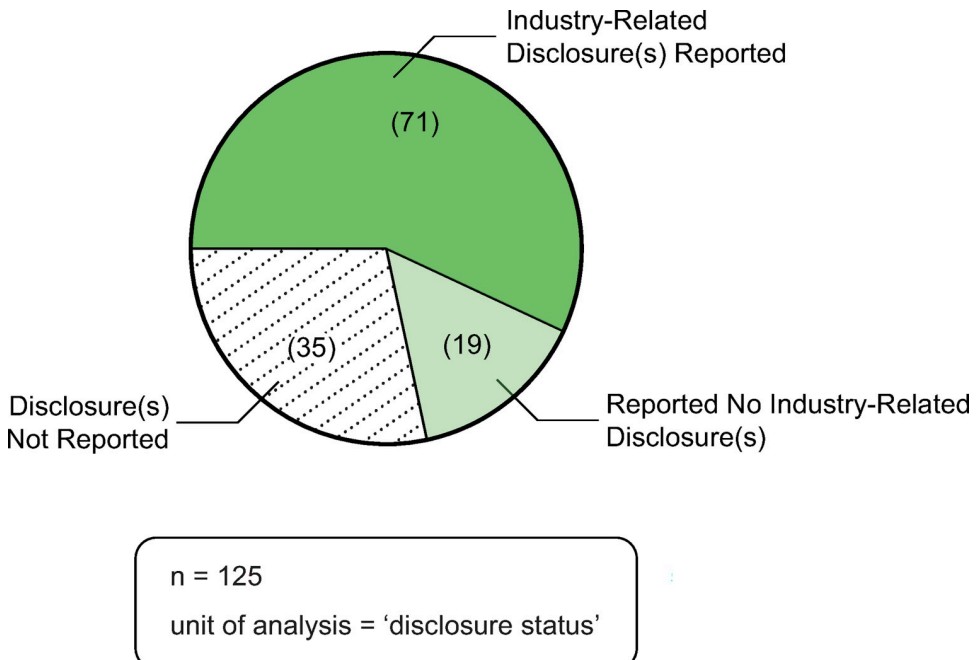

**Fig 9. Disclosures of interest.** Source: Microsoft Excel was used to facilitate data analysis; Adobe Illustrator was used to facilitate data visualization.

One publication reported no affiliation information. Of those publications that explicitly reported one or more industry-related author affiliations: 53 publications reported genomic- and/or diagnostic-related affiliations; 30 publications reported consulting- and/or research-related affiliations; and 10 publications reported both genomic- and/or diagnostic-related and consulting- and/or research-related affiliations.

## Discussion

Together, these bibliometric and funding analyses provide a detailed analysis of decision impact studies in genomic medicine research, with a focus on industry involvement in evidence production. This study used a significant number of data points and analyzed the data from numerous perspectives, including investigating industry involvement through author affiliations, proprietary test focus (i.e., disease-type), collaborations based on proprietary products, funding sources, and disclosures of interest. Our previously conducted scoping review shed light on decision impact studies in genomic medicine research in cancer care and found that these studies are predominantly produced to report indicators of clinical utility of proprietary genomic tests and to support obtaining both private and public coverage or reimbursement [4]. The results of this study extend our understanding of this new research and show that decision impact studies are industry-conceived, -funded and -produced evidence. These conclusions are illustrated through key findings regarding the prevalence of industry funding and financial involvement of these studies, authors, collaborations and 'invisible colleges', and the dominance of test producers' involvement in decision impact studies.

The findings on the extent of industry funding in decision impact studies is particularly illuminating. This trend is not unique to genomic medicine, as similar trends have been identified in other health-related industries, such as pharmaceuticals [9], alcohol [11, 12] and tobacco [15]. Our study adds to this body of research by explicating the extent of industry funding in

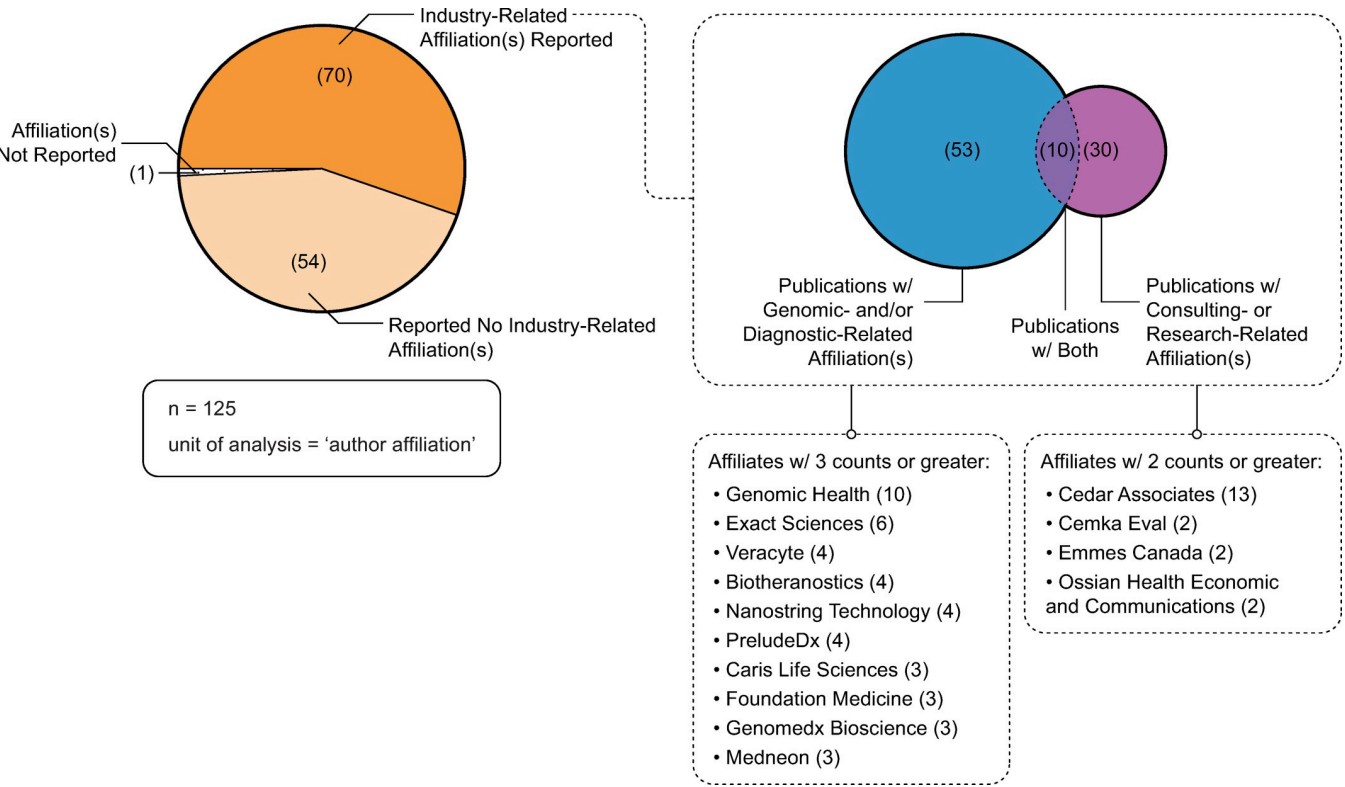

**Fig 10. Industry-related affiliations, genomic- and diagnostic-related and consulting- and research organization-related.** Source: Microsoft Excel was used to facilitate data analysis; Adobe Illustrator was used to facilitate data visualization.

decision impact studies in genomic medicine research. Of those publications that reported funding sources (n = 61), over 90% of the sources reported were industry-related (n = 56), 87% of which were genomic- and/or diagnostic-related funding sources (n = 53). In addition, of those publications that reported industry-related affiliations (n = 70), 76% of publications of had an author with a genomic- and/or diagnostic-related affiliation (n = 53)—almost 20% of which also had an author with a consulting- or research-related affiliation. These results provide insight into industry funding in the production of decision impact studies. In addition, disclosures of interest are an important, but often poorly reported, aspect of the funding of scientific publications [11, 18, 44]. Even with the caveats of missing and inaccurate disclosures, our analysis explicates the depth and complexity of industry's involvement with these studies. Similar to pharmaceutical research [9], despite the lack of disclosures, 57% of the publications in our study reported industry-related disclosures of interest. Of note, 28% of publications did not include information about authors' disclosures of interest. Scholars have raised concerns about the pervasive issue of missing, incomplete and inaccurate reporting of disclosures of interest [44, 45] and have made calls to enhance transparency by asking journals to enforce reporting and to adopt standard nomenclature [18]. To ameliorate some of these issues we collected all of the funding data manually (directly from publications, rather than relying on the metadata) and were able to demonstrate that the majority of included decision impact studies reported industry related disclosures. With 28% publications not reporting disclosures of interest these data demonstrate the prevalence of industry funding in the production of decision impact studies.

Our results illuminate the prevalence of industry-affiliated 'invisible colleges' of author networks [25, 28–31] in decision impact studies. More than half of the included publications

reported that one or more authors were affiliated with industry (and most commonly affiliated with the company that produced the test evaluated in the study). Our author collaboration analyses revealed specific and focused collaboration networks, clearly established along proprietary product and company lines. The 23 collaborations identified, and the focus of the research conducted by these collaborations, demonstrates the use of decision impact studies by the companies developing these proprietary tests. Of note, some authors were part of two different networks, demonstrating the spread of the use of decision impact studies between different cancer-types and genomic tests. These cross-product collaborations demonstrate the perceived utility of decision impact studies and illustrate the effectiveness of 'invisible colleges' at building and sharing knowledge. These findings align with Demortain's application of 'invisible colleges' as these networks appear to be established to shape regulatory outcomes, specifically, to navigate coverage and reimbursement processes [31, 46]. The author collaboration network revealed in our study supports Demortain's assertion that the invisible college develops common concepts and standards [31], as the authors in our study are producing decision impact studies, studies shown to have a narrow focus on reporting outcomes of clinical utility for these tests [4]. The networks lend authority to the outputs, aiming to influence decision-making and working to gradually normalize industry involvement in the development and dissemination of these tests [25]. The circuit of authors who produce and disseminate decision impact studies constitute the invisible college and, in turn, become, "a constitutive part of the broader regulatory regime" [46 (p.2)] to produce evidence of clinical utility for these products. The high prevalence of authors with industry affiliations aligns with observations that, increasingly, industry is embedding itself throughout the research process [14, 47]. Also of note is the fact that 24% of studies reported at least one author affiliated with a CRO. The growing prevalence of CROs in medical technology research and development has been noted by scholars [14, 47] and requires further research to understand this shift in research production and its implications for the quality of evidence produced.

The final key finding highlights industry's dominant role in the creation, conduct and dissemination of decision impact studies in genomic medicine research. The first published use of the phrase 'decision impact study' was in 2010. Beginning in 2010, publications concentrated on breast cancer genomic assays—as demonstrated by the numerous, breast cancer assay focused-conference abstracts that were published between 2010 and 2012. Indeed, the 15 conference abstracts published in this short time window, all for Oncotype Dx, demonstrate that Genomic Health and collaborating researchers were likely to be responsible for the introduction of the 'decision impact study' concept and phrase. The growth of publications reporting on or about decision impact studies since then shows an evolution in the production of this new form of evidence; publications expanded to proprietary tests other than Oncotype Dx in 2013, though still focused on breast cancer; also, publications extended to other types of cancer, and then to diseases beyond cancer (lung diseases and rheumatoid arthritis). These findings highlight the scale of industry's participation in the production of scientific evidence [14, 24], and further demonstrate how private firms play a central role in the creation of new forms of knowledge. The analysis of title topics further demonstrates industry's central position in this new field. Interestingly, the fact that two proprietary terms—'Oncotype Dx' and 'Recurrence Score'—were top title terms, highlights the dominance of proprietary products in the focus of decision impact studies. For example, a publication involving Oncotype Dx, but not conducted by industry, often referred to the genomic test as the '21-gene assay' and not by its brand name 'Oncotype Dx'. The fact that a brand name is in the title of a journal article illustrates industry's involvement in the production of these studies and is a clear marketing technique, confirming the value of brand recognition. In addition, decision-making terms, including 'clinical utility', also figured prominently in the publication titles. This explicit

labeling calls attention to the expressed purpose of decision impact studies to provide evidence of clinical utility of these proprietary tests.

## Strengths and limitations

This study has many strengths. Myriad steps were taken to ensure that rigour was upheld throughout our study, improving the validity and reliability of our results: multiple researchers were involved in verifying that data were accurately collected; decisions concerning the cleaning of data were discussed collectively and recorded in a data cleaning log; advice regarding data cleaning best practices was solicited from experienced methodologists; the software creators were contacted when bugs in the software were suspected; and, the results reflected in the data visualizations were cross-referenced with the original datasets. Data visualizations produced by the software were enhanced with further analysis or to improve interpretation of results. In addition, we manually added publications that were in scope, but not available through WoS. We also decided, due to the incomplete nature of the electronic metadata available, to do the funding analysis manually. As well, a more conservative approach was taken for the funding analysis to ensure that the analyses were not artificially inflated; care was taken to exclude publications that reported empirical results from the same decision impact study.

This study has some limitations. Due to resource constraints, we only included English-language publications. Our study is an important contribution to the field, but with the assumed caveat of the quality of data available: we were only able to analyze bibliometric metadata reported in publications and databases, and as discussed above we cannot assume these data are fulsome or entirely accurate. We acknowledge that the lack of data normalization for funding acknowledgments in electronic databases is a limitation [17]. Our rigorous data cleaning and manual data collection processes mitigated some of these limitations.

## Directions for future research

Understanding industry's involvement in decision impact studies is critical to understanding their intended purpose and the motivations behind their production. While this study significantly contributes to the foundational information we are producing on decision impact studies, it also highlights important topics for further inquiry in the field: *How effective are decision impact studies in supporting industry-favourable coverage and reimbursement decisions*? *What are authors' and researchers' perspectives on the increasing role of CROs in scientific research production*? *What motivates genomic medicine researchers to align themselves with industry (i. e., to be part of the industry-directed scientific evidence production system*?*)

## Conclusions

The findings of this study provide a rigorous and comprehensive bibliometric mapping of decision impact studies, a new and expanding type of research in the field of genomic medicine. Our study provides insights into key authors, relationships and networks, affiliates and industries and funding relationships. The results provide important insights on 'invisible colleges' of collaboration, industry's role in evidence production, the prevalence of industry funding and involvement and the growing trend of industry-supported evidence production for coverage and reimbursement.

## Supporting information

**S1 Appendix. Full electronic search strategy for scopus database.**
(DOCX)

**S2 Appendix. Metadata used in the bibliometric analysis.**
(XLSX)

**S3 Appendix. Metadata used in the funding analysis.**
(XLSX)

**S4 Appendix. R-code used to identify title topics.**
(DOCX)

**S1 Table. Table of included publications.**
(DOCX)

## Acknowledgments

The authors would like to thank Samer Ghazi, Dr. Lena Saleh, Dr. Duncan Moore, and Helen Valkanas for their valuable assistance.

## Author Contributions

**Conceptualization:** Gillian Parker, Stuart Hogarth, Fiona A. Miller.

**Data curation:** Gillian Parker, Sarah Hunter.

**Formal analysis:** Gillian Parker, Sarah Hunter.

**Funding acquisition:** Fiona A. Miller.

**Investigation:** Gillian Parker, Sarah Hunter, Fiona A. Miller.

**Methodology:** Gillian Parker, Sarah Hunter, Stuart Hogarth, Fiona A. Miller.

**Project administration:** Gillian Parker, Fiona A. Miller.

**Resources:** Fiona A. Miller.

**Software:** Gillian Parker, Sarah Hunter.

**Supervision:** Gillian Parker, Stuart Hogarth, Fiona A. Miller.

**Validation:** Gillian Parker, Sarah Hunter, Stuart Hogarth, Fiona A. Miller.

**Visualization:** Gillian Parker, Sarah Hunter.

**Writing – original draft:** Gillian Parker, Sarah Hunter.

**Writing – review & editing:** Gillian Parker, Sarah Hunter, Stuart Hogarth, Fiona A. Miller.

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
