## [Decision Letter · Decision Letter 0]

28 Mar 2023

PONE-D-23-02332Industry involvement in evidence production for genomic medicine: A bibliometric and funding analysis of decision impact studiesPLOS ONE

Dear Dr. Parker,

Thank you for submitting your manuscript to PLOS ONE. After careful consideration, we feel that it has merit but does not fully meet PLOS ONE’s publication criteria as it currently stands. Therefore, we invite you to submit a revised version of the manuscript that addresses the points raised during the review process.

Please add the missing information and modify the paper according to the Reviewers' suggestions.

Especially please develop Research Questions, and add the highlights and implications as well as future works. Please provide the missing R code and modify the sections in the Introduction.

The required details are attached in the revisions.

We look forward to receiving your revised manuscript.

Kind regards,

Agnieszka Konys, Ph.D.

Academic Editor

PLOS ONE

Reviewers' comments:

Reviewer's Responses to Questions

**Comments to the Author**

1. Is the manuscript technically sound, and do the data support the conclusions?

Reviewer #1: Yes

Reviewer #2: Partly

2. Has the statistical analysis been performed appropriately and rigorously? 

Reviewer #1: Yes

Reviewer #2: Yes

3. Have the authors made all data underlying the findings in their manuscript fully available?

Reviewer #1: Yes

Reviewer #2: No

4. Is the manuscript presented in an intelligible fashion and written in standard English?

Reviewer #1: Yes

Reviewer #2: Yes

5. Review Comments to the Author

Reviewer #1: I have following obseravtions on the paper

1. I would suggest authors to create/develop Research questions that this review is aimed to answer. Authro can refer the following papers for development of the RQs

Kumar, S., Sharma, D., Rao, S. et al. Past, present, and future of sustainable finance: insights from big data analytics through machine learning of scholarly research. Ann Oper Res (2022). https://doi.org/10.1007/s10479-021-04410-8

Cumming, D., Jindal, V., Kumar, S., & Pandey, N. (2023). Mergers and Acquisitions Research in Finance and Accounting: Past, Present, and Future. European Financial Management.

2. Once you developed RQs you can use these RQs for the organisng your results and presneting them so that readers can be benefited. Also please add indigshts to each analysis what it means before your report what you found in anlysis.

3. Conclusion can be strengthened by highlighting major implications of findings

4. What next ? Based on your findings can you propose some future reseach questions that future scholars can use to pursue research in the filed.

Wish you luck.

Reviewer #2: Tell readers and provide code in R for drawing Figure 2 and 8-10.

Study data should provided with a textfile that meeting the study under the biblioshiny app in biblimetrics

Two sections in introduciton and discussions should be rewritten for readers to pursuit the authors' tempo in this study.

6. PLOS authors have the option to publish the peer review history of their article (what does this mean?). If published, this will include your full peer review and any attached files.

Reviewer #1: No

Reviewer #2: No

---

## [Author Response · Author response to Decision Letter 0]

9 Apr 2023

Please refer to responses included in attached file 'Response to Reviewers.pdf'.

---

## [Editor Report · Decision Letter 1]

16 Apr 2023

Industry involvement in evidence production for genomic medicine: A bibliometric and funding analysis of decision impact studies

PONE-D-23-02332R1

Dear Dr. Parker,

We’re pleased to inform you that your manuscript has been judged scientifically suitable for publication and will be formally accepted for publication once it meets all outstanding technical requirements.

Kind regards,

Agnieszka Konys, Ph.D.

Academic Editor

PLOS ONE
---

## [Editor Report · Acceptance letter]

19 Apr 2023

PONE-D-23-02332R1 

Industry involvement in evidence production for genomic medicine: A bibliometric and funding analysis of decision impact studies 

Dear Dr. Miller:

I'm pleased to inform you that your manuscript has been deemed suitable for publication in PLOS ONE. Congratulations! Your manuscript is now with our production department. 

Kind regards, 

on behalf of

Dr. Agnieszka Konys 

Academic Editor

PLOS ONE